# Challenges facing interdisciplinary researchers: Findings from a professional development workshop

**Kristy L. Daniel**[1], **Myra McConnell**[2], **Anita Schuchardt**[2], **Melanie E. Peffer**[3] *

**1** Department of Biology, Texas State University, San Marcos, Texas, United States of America, **2** Biology Teaching and Learning, University of Minnesota, Minneapolis, Minnesota, United States of America, **3** Health Professions Residential Academic Program & Institute of Cognitive Science, University of Colorado Boulder, Boulder, Colorado, United States of America

* melanie.peffer@colorado.edu

**Data Availability Statement:** All relevant data are within the paper and its Supporting information files.

## Abstract

Interdisciplinary research is the synergistic combination of two or more disciplines to achieve one research objective. Current research highlights the importance of interdisciplinary research in science education, particularly between educational experts within a particular science discipline (discipline-based education researchers) and those who study human learning in a more general sense (learning scientists). However, this type of interdisciplinary research is not common and little empirical evidence exists that identifies barriers and possible solutions. We hosted a pre-conference workshop for Discipline-Based Educational Researchers and Learning Scientists designed to support interdisciplinary collaborations. We collected evidence during our workshop regarding barriers to interdisciplinary collaborations in science education, perceptions of perceived cohesion in participants' home university departments and professional communities, and the impact of our workshop on fostering new connections. Based on participants' responses, we identified three categories of barriers, *Disciplinary Differences*, *Professional Integration*, and *Collaborative Practice*. Using a post-conference survey, we found an inverse pattern in perceived cohesion to home departments compared to self-identified professional communities. Additionally, we found that after the workshop participants reported increased connections across disciplines. Our results provide empirical evidence regarding challenges to interdisciplinary research in science education and suggest that small professional development workshops have the potential for facilitating durable interdisciplinary networks where participants feel a sense of belonging not always available in their home departments.

## Introduction

Science education research, particularly at the higher education level, is amid a major shift. While researchers have documented effective pedagogical practices, questions still remain as to *why* it works and *who* it works for. Some have referred to this shift from *what* to *why* and for *whom* as the "second-generation" of discipline-based education research (DBER) [1].

**Funding:** KLD, MM, AS, MF: This work was funded by the National Science Foundation grant (DUE #2017278). https://www.nsf.gov/ The funders had no role in study design, data collection and analysis, decision to publish, or preparation of the manuscript.

**Competing interests:** The authors have declared that no competing interests exist.

Discipline-based educational researchers (DBERs) have strong content knowledge within their discipline and largely investigate educational research questions isolated within that content field. In contrast, learning scientists (LSs) have more expertise in human learning as they explore educational research through broader questions about learning [2]. As part of the shift to the second-generation of DBER, the National Research Council has called for the inclusion of perspectives from other disciplines, in particular learning sciences (LS), to enhance DBER [e.g., 3–6]. In particular, there is a need for interdisciplinary collaborations where perspectives from each field are intentionally combined to generate novel insights into learning in science classrooms.

Although many advocate for interdisciplinary research in science education to enhance our understanding of how people learn science [1–8], there are also barriers to interdisciplinary research in science education. These range from differences in research methodologies and practices to disciplinary siloing. Prior work by several research groups identified these and other barriers, but only provided theoretical or personal perspectives [e.g., 2, 6, 9]. There is a lack of empirical studies on barriers (and also possible solutions) to foster interdisciplinary collaborations between LSs and DBERs. Therefore, in order to develop evidence-based best practices for facilitating interdisciplinary collaborations, it is necessary to generate empirical evidence.

Building on these insights from theoretical work, we organized a two-day virtual workshop for DBERs and LSs interested in science education. Our goal with the workshop was to both connect researchers across disciplines and facilitate the development of the skills necessary to engage in interdisciplinary research, but also to generate empirical evidence regarding barriers.

## Theoretical framework

We classify research approaches into the following five categories: intradisciplinary, multidisciplinary, cross-disciplinary, transdisciplinary, and interdisciplinary [10–12]. Within intradisciplinary research, investigators use norms within a single discipline to address research questions applicable to that discipline. Multidisciplinary research draws on knowledge from different disciplines but stays within their borders providing various perspectives to address complex, real-world problems. Cross-disciplinary research involves using single disciplinary methods and assumptions to cross borders to address questions about a topic outside the scope of the discipline without any integration from other disciplines. And transdisciplinary research occurs when ideas from a discipline(s) offer insights that transcend the discipline's traditional borders. Interdisciplinary research integrates research methods, knowledge, assumptions, and frameworks from separate disciplines to address a shared research question.

For our study, we focused on interdisciplinary research. The key distinguishing characteristic from other forms of collaborative research is that interdisciplinary research relies on a synergistic combination of discipline-oriented viewpoints to solve a common problem. In particular, we were interested in interdisciplinary collaborations between two multidisciplinary groups, the human expertise of LSs and the content and pedagogical expertise of DBERs. Interdisciplinary research is a valuable avenue for gaining novel insights into how people learn and understand science [2, 3, 6].

Individual expertise is more than a collection of facts from a discipline; it also reflects what a field values, considers problem spaces and worthwhile problems to pursue, what constitutes data, and how that data is collected and disseminated [13]. Therefore, although interdisciplinary collaborations are powerful because they unite disparate disciplines, the very nature of being interdisciplinary requires overcoming an array of barriers. For example, within an

interdisciplinary collaboration, each investigator is an expert in one disciplinary culture (e.g., DBER) while a non-expert, or novice, in the other disciplinary culture (e.g., LS). Experts and novices conceptualize and approach problems differently as a factor of their experience, contributing to additional time constraints and the necessity of ongoing communication [13]. As such, interdisciplinary research teams must learn how to communicate and work together to cross cultural borders.

Border crossing has potential advantages such as greater learning and research productivity for participants. Four mechanisms that explain how participants learn at the border of two cultures have been identified: *Identification*, *Coordination*, *Reflection*, and *Transformation* [14]. *Identification* refers to the process of defining how practices within distinct cultures are similar and different from one another. *Coordination* is the mechanism by which members of different cultures figure out how to allow distinct practices to work effectively together and requires clear communication between the members of the two cultures. *Reflection* is the process of figuring out how and why practices are different and consequently learning something new about those practices. *Transformation* captures changes in practices, potentially the creation of a new interdisciplinary collaborative practice. This transformation often comes about as the result of a confrontation between two distinct practices from the different cultures that must be reconciled to allow work to progress. These mechanisms of learning that occur while border crossing can account for the synergy and creativity that is often found in interdisciplinary work [6]. We assume that practices facilitating these four mechanisms can assist with overcoming existing barriers and will support connections to build interdisciplinary networks.

Although crossing cultural borders to perform interdisciplinary research and form interdisciplinary communities and collaborations is advantageous, it can carry risks for early-career researchers [15]. Because interdisciplinary research often requires additional time to account for communication of cultural differences, productivity may take longer than with intradisciplinary pursuits. Time is needed to find interested collaborators, generate a shared language and goal, and read unfamiliar literature [15–18]. As such, early-career researchers may avoid interdisciplinary pursuits further siloing departments and cultures. Therefore, interdisciplinary faculty members often identify as "disciplinary outcasts" resisting socialization into a single discipline to some extent or managing to find a way to navigate the border between two cultures [19]. Faculty that cross borders may be seen by themselves or others as not truly belonging to either disciplinary community [14]. This lack of community can lead to feelings of isolation [20, 21] and potentially lead the interdisciplinary researcher to have low perceived cohesion with their department [22]. Consequently, the formation of interdisciplinary communities can aid the development of an interdisciplinary culture which supports and socializes new members and promotes crossing borders.

## Theorized barriers to interdisciplinary research in science education

Although many advocate for interdisciplinary research between LSs and DBERs [1–8], and the two communities have the shared goal of understanding learning, interdisciplinary research in science education is limited. The lack of interdisciplinary research between LSs and DBERs can be traced to a variety of factors, including disciplinary differences in research practices and culture, disciplinary siloing, and risk-reward ratio for early career researchers. As a result of these barriers, interdisciplinary research between LSs and DBERs tends to not occur organically and consequently, targeted measures such as the intentional development of interdisciplinary communities or dedicated research seminars are important to facilitate interdisciplinary research.

As described above, disciplines serve as cultural structures that transmit and preserve creative products and knowledge [23–25]. As cultural structures, each discipline has its own values, beliefs, norms, activities, and practices [26]. Although valuable for differentiating disciplines, these differences can present problems in collaborations that cross disciplinary boundaries. For example, differences in ways of describing a research problem, approach to that research problem, and how research is discussed are all practice barriers [6, 13, 16, 17, 27]. Language use between the disciplines can create confounding issues; the term "experiment" can vary in meaning based on disciplinary use [9]. Within DBER, to experiment typically requires manipulation by the investigator, whereas in LS to experiment generally refers to any comparative study or a systematic way to investigate something with an intervention group only. These disciplinary cultures have developed over time to have different "traditions, social organization, reward systems, and especially an offering of professional status and dignity" [28, p.41]. To engage in interdisciplinary research, faculty need to learn how to communicate across two or more disciplinary cultures. The process of working between two cultures has been referred to as border crossing [29]. Border crossing involves changing behavior to match the expectations of the new group. Some border crossings are easier than others. For instance, many people move between their home and work culture with little difficulty. However, other crossings can be more challenging, particularly if there are institutional or systemic barriers to overcome.

Institutions tend to divide higher education into disciplinary silos [15]. Disciplines can also be referred to as "tribes," each with their own territories [30]. The definition of what constitutes knowledge and evidence is a cultural construct determined by discipline and thus each discipline has developed its own standards for what constitutes high-quality work [15, 31]. Most university research faculty have been socialized into the culture of their discipline through a process of legitimate peripheral participation where junior members pick up the values, beliefs, and practices of a culture through their participation as members of a department within a university and their interactions with established members of the disciplinary culture [15, 32]. Faculty members' identities are thus forged by the disciplinary culture which affects assumptions about what constitutes knowledge, how to interact within the discipline (professionally and socially), publication patterns, how to achieve status, and what it means to be successful professionally [23]. As a result, there can exist animosity between differing kinds of research [6, 18, 27]. For example, LS often comes from the "soft" sciences, tending to be housed with the College of Education or College of Liberal Arts. Whereas, DBER often comes from the "hard" sciences, tending to be housed within a researcher's home content department, typically in a College of Science. Part of this animosity stems from a fundamental misunderstanding of the diversity of research practices [6]. What constitutes rigorous practice in one discipline may not be as accepted by a different discipline because the approach may not adhere to norms within the discipline [2].

Disciplinary silos extending beyond university settings are encountered within funding agencies and dissemination outlets. Institutional funding, rewards, and other resources are often distributed by university departments [15]. Reviewers of grant proposals typically apply the standards of knowledge for their discipline, not adjusting expectations for the interdisciplinary nature of a proposal [33]. As a result, funding supporting interdisciplinary research can be more difficult to receive [34]. The predominance of specialized journals and conferences also contributes to professional disciplinary siloing [16, 17]. For example, researchers tend to publish findings in journals and attend conferences that are specific to their discipline, and generally do not read papers or interact with many researchers outside of their discipline [2]. This siloing is further reflected in citations found within publications, as people tend to cite within their own discipline [17]. Consequently, interdisciplinary researchers may have difficulty finding a suitable outlet to publish their work. If researchers choose to pursue

publishing in journals outside of their primary disciplines, they need to be familiar with and adapt to different writing styles and evidence standards. Therefore, work that falls outside the bounds of the cultural expectations can be difficult both to publish and get funded [15, 33, 34]. Furthermore, interdisciplinary journals also tend to be ranked lower than disciplinary-specific journals in terms of their overall impact [35].

The lack of well-established interdisciplinary journals not only contributes to challenges with disseminating knowledge, but can also limit potential interdisciplinary tenure-track faculty members who need to publish in reputable journals to be promoted [16]. Disciplinary siloing creates a greater risk for interdisciplinary faculty within their department both in terms of receiving resources and in terms of career advancement [15, 33, 35]. For example, there are differences in assumptions about what constitutes authorship and the meaning of authorship order. Senior authors may be either found in the last position or order is based on percent contribution in DBER papers, but senior authors are typically listed first in LS papers. This is a difference that often has implications in promotion discussions and tenure for university faculty [27].

Interdisciplinary research is widely regarded as crucial for advancing our understanding of how people learn science [1–8]. Although some papers describe general observations made by researchers, these articles have only provided theoretical perspectives [e.g., 2, 6, 9]. Furthermore, while there have been limited empirical studies on the contributions of interdisciplinary as opposed to single-discipline research [e.g., 17, 36], there is a scarcity of empirical studies identifying the barriers to interdisciplinary research between LSs and DBERs and strategies for fostering interdisciplinary connections.

## Methods

### Context of current study

Using insights from theoretical work, our interdisciplinary research team (two researchers classified as DBERs and two LSs) organized an NSF-funded (DUE #2017278) two-day virtual workshop in conjunction with a large international conference, to connect researchers and equip them with skills to engage in interdisciplinary research. To combat the lack of empirical evidence on LS and DBER interdisciplinary research barriers and strategies, we collected data regarding barriers and solutions for overcoming those barriers among our participants (Identification and Coordination), perceived cohesion within the professional research community (Reflection), and formation of new collaborations post-workshop (Transformation). In this paper, we describe implications for how our findings can be leveraged to create needed support for building interdisciplinary communities of practice and provide recommendations for future interdisciplinary research or professional development to facilitate interdisciplinary research. The following research questions guided our study:

1. What are participant-identified challenges of interdisciplinary research?

2. To what extent do participants' report perceived cohesion to their professional communities and home academic departments?

3. What meaningful connections arose between workshop participants?

### Workshop participant activities

We held a virtual workshop in conjunction with a large international conference. Participants (n = 33) represented four countries (USA [12 from the New England region, two from the Southeastern region, eight from the Midwest region, one from the Northwestern region, and

five from the Southwestern region], Canada [two], India [one], UK [two]), including 18 trainees (graduate students or postdoctoral scholars), six mid-career faculty, and nine early career researchers. There were 20 women and 13 men. The participants represented a variety of disciplines with 15 DBERs from physics, chemistry, biology, mathematics, and geology education and 18 LSs from cognitive psychology, educational psychology, instructional design, and general science education. Furthermore, most participants represented their own institution with only four institutions having up to three participants in attendance. Participants were recruited through conference registration, workshop announcements through professional listservs, and by personal invitations from the workshop leaders. To maintain the confidentiality of participants, identifiers were removed, and all research was conducted in accordance with Institutional Review Board guidelines (Approval #20–0077). We received written informed consent from each participant.

During the workshop, participants engaged in four core activities designed to foster community and to engage participants in the learning mechanisms (Identification, Coordination, Reflection, and Transformation) associated with border crossing [14]. These activities were: 1) Interactive discussion of interdisciplinarity in biology education and strategies for collaboration, 2) Reflection on collaboration as a way to cross borders; 3) Panel showcase of current interdisciplinary research, and 4) Presentation of participants' research during a virtual poster and networking session.

**Activity 1: Discussion of interdisciplinary differences.** We assigned participants into interdisciplinary groups (of three-four individuals each) to explore a de-identified educational data set that could be addressed through multiple lenses. Data in this set included: 1) student survey data on perceptions of equations commonly used in biology, 2) students' step-by-step solutions to a quantitative biology problem, 3) students' self-perceptions (e.g., science identity, persistence on task, self-efficacy on science processes, academic self-efficacy), 4) recordings of instructors teaching an equation, and 5) instructor perceptions of the role of mathematics in biology. We chose this data set because mathematics in science is a practice that spans scientific disciplines, includes both attitudinal and performance measures, and integrates data from both instructors and students. Individuals generated and shared three research questions they were most interested in exploring within their group. During group discussions, participants considered potential analytical methods that could be applied to address each question. Group conversations surfaced differences in research interests, analytical methods, and interpretation of data among individual researchers (to promote Identification). Each group selected one prioritized research question representing an interdisciplinary approach to investigate the sample data set and presented their rationale to the full workshop group using Google slides (to promote Coordination).

**Activity 2: Reflection on collaboration as a way to cross borders.** After the group sharing of interdisciplinary research questions from the collaboration activity, participants rejoined their breakout groups and engaged in a think-pair-share reflection. Participants initially reflected upon disciplinary differences they observed during the collaboration activity and considered ideas for bridging approaches in order to cross borders (to promote Reflection). Then, participants took turns sharing their reflections within their breakout groups to generate a list of barriers they identified as well as potential strategies they considered for overcoming the challenges on a collaborative Google slide. Finally, each group shared their reflections with the full workshop group while the research team facilitated the discussion by generating a consensus list of strategies for facilitating successful interdisciplinary collaboration.

**Activity 3: Panel to showcase successful interdisciplinary research.** Four researchers involved in prior interdisciplinary research projects led a panel discussion about their prior

experiences. The panelists represented advanced career researchers that have engaged in inter-disciplinary research collaborations for the majority of their time in higher education. Each panelist highlighted past challenges, successes, and lessons learned about interdisciplinary collaborations. Specifically, the panelist addressed issues such as how they found collaborators outside of their home discipline, handling departmental expectations regarding productivity, and the importance of belonging to a supportive professional community. Following the panel presentation, participants engaged in a virtual question and answer session with the panelists to explore additional issues that had arisen during the prior reflection period.

**Activity 4: Promote networking opportunities and formation of new collaborations.** During the second day of the workshop, attendees presented their own research via concurrent virtual poster sessions. Posters were presented during six concurrent sessions to maximize opportunities for attendees to both present their posters and view other presenters' posters. We scheduled similarly themed posters to present at the same time to encourage participants to attend posters they might not otherwise have considered attending. We purposefully designed the session in this manner to promote the cross-pollination of new ideas and perspectives across disciplinary boundaries and allow for ample networking interactions (to promote Transformation). After the poster session, we regrouped for a brief discussion of perceived outcomes from the workshop and future endeavors.

## Data sources

Data was collected from three sources: surveys of participants, field notes from workshop observations, and artifacts from the workshop including chat transcripts and the Google slides. Additionally, during the workshop, video-recording occurred for full group and breakout discussions, activities, and poster presentations.

Participants were surveyed at three different time points (T1, prior to the workshop; T2, one-week post; and T3, one-month post). The goal of these surveys was to determine if the workshop's objectives had been met and to triangulate interpretations of qualitative data collected during the workshop. In the initial questionnaire (T1) participants provided demographic information on their experience as a researcher, identified their primary and secondary (if relevant) research discipline, and justified their motivations for participating in the workshop. In the second questionnaire (T2), we assessed participant-reported learning outcomes by using open-ended prompts asking participants to share what they learned and valued from their participation, who they had interacted with prior to and after involvement in the workshop, and suggestions for future events. In the final questionnaire (T3), we focused on documenting participant networks among workshop attendees, reported interest in future types of networking and interdisciplinary professional development activities, and Likert-type follow-up responses about themes addressed during the workshop (to measure Transformation).

We measured participant connection in two ways: perceived cohesion and networking. We used the first measurement of perceived cohesion, which data collection occurred at T3, to validate the theme of difficulty with professional integration that had emerged from the qualitative analysis. The Perceived Cohesion Scale [37] contains two constructions, belonging and morale, and we used this instrument to examine participants' perceived cohesion to both their home departments and their self-identified professional society. The Perceived Cohesion Scale measures sense of belonging (the perception of inclusion through recognized affiliation, $\alpha =$ 0.95) and feelings of morale (emotional response in relation to involvement, $\alpha = 0.95$) using a 6-point Likert-type scale [22, 37]. In order to capture evidence of participant connections through their networking, we examined the extent to which the workshop had made progress

towards improving interdisciplinary community interactions. On the T3 survey, participants used a provided list to self-report which workshop attendees they interacted with as potential research collaborators before and after the workshop. We defined "interacted with" as having a meaningful professional relationship, such as collaborating on a research project or serving on a committee together.

## Data analysis

Rather than approaching the data with predetermined themes in mind, we used an inductive approach and let the data lead us to identify themes. We created descriptive codes to analyze data across all qualitative sources [38]. We used a constant comparative approach across data sources during our initial coding cycle to prevent potentially missed challenges. To increase the reliability of our coding, we utilized investigator triangulation [39]. At least two members of our research team individually coded each data source and then compared codes for accuracy. When discrepancies arose between researchers, differences were discussed until a consensus over conflicting ideas was reached and a final coding was agreed upon. For example, some of our initial codes about challenges included the following: struggle with understanding jargon, issues finding colleagues with shared research interests, problems finding interdisciplinary research funding. By utilizing our inductive coding approach in conjunction with sensitizing concepts derived from prior research about collaboration challenges, we were able to group like codes, making sure the full research team agreed with all groupings. Lastly, we reviewed our analysis to identify evidence of the three emergent themes encompassing major challenges of interdisciplinary researchers from data we collected during our workshop: 1) *Disciplinary Differences*, 2) *Professional Integration*, and 3) *Collaborative Practicalities*. We used the one-month post (T3) workshop survey to triangulate our prior data interpretations and improve the trustworthiness of our results. In our findings, we provide rich descriptions of the emergent themes along with supporting quotes from the data as evidence of the identified challenges.

We calculated a separate average score for morale and feelings of belonging for both a professional community and home department for each participant. We used paired t-tests to compare morale and feelings of belonging to determine significance.

We used participants' lists of who they interacted with from the workshop at T3 to create a visualization of the social network before and after the workshop using the ForceAtlas2 layout algorithm in Gephi v0.9.2. We used the following criteria for inclusion in the network analysis: participation in the workshop and inclusion on the list of people that survey respondents could select from as contacts.

## Results

Three themes emerged from the challenges that participants identified: *Disciplinary Differences*, *Professional Integration*, and *Collaborative Practice*. These themes were not mutually exclusive, and the challenges often built upon each other, which we will discuss more below. Additionally, we found that the perceived cohesion of the interdisciplinary research participants varied when we looked at the factors of sense of belonging to a community and their feelings of morale towards their involvement. We reported average means of participant perceived cohesion to identified professional communities and home academic departments. We also explored collaboration networks across our participants to better understand the communication and connections that grew after participating in the workshop. We reported participant connections through network analysis.

## Identified challenges

**Disciplinary differences.**   Participants struggled with disciplinary differences for two main reasons: communication discrepancies and conflicting paradigms. Participants noted language inconsistencies across researchers from varying disciplines that impacted communication. As one participant stated, a challenge arose during communication because of, "The jargon that we use." When using specific terms, "We might not think it's jargon," but the lack of understanding of some terms by collaborators created an unanticipated need to provide definitions.

Participants also noted that they had to resolve approaching research from differing methodological/theoretical approaches and standards due to conflicting disciplinary paradigms. Participants said that a valuable starting point to overcoming conflicting paradigms was to first clarify backgrounds. According to one of the attendees, "One of [the] strategies for bridging differences was sharing our different perspectives and being more explicit about the assumptions that we held and how we were defining different things that we were talking about." Another participant also iterated this idea,

> It is important to have conversations among researchers that help 1) define a shared problem and then 2) unpack the different theories of action that come out of different disciplines [that] can help us to 'try on each other's hats' which pushes a broader and more critical approach to undertaking the work.

The differences in professional language and theoretical frameworks employed created communication challenges when initially developing interdisciplinary research questions.

**Professional integration.**   Participants struggled with integrating themselves into professional communities, particularly their home departments as interdisciplinary researchers. As one participant stated,

> I don't think that I am seen by the physics education research community as a central member. I don't think I am seen by the learning science community as a central member. I don't think I am seen as a central member of any community.

Additionally, the interdisciplinary researcher participants struggled with forming their professional identity. Another participant stated that prior to the workshop,

> I would have considered myself an interdisciplinary researcher, but I am seeing that I am really one type of interdisciplinary researcher and that has given me an identity that is fairly fixed as opposed to some who see their identity as fluid within this space of interdisciplinary.

Participants also identified a need for clarifying expectations with peer colleagues who may not fully understand the professional norms and expectations associated with interdisciplinary research activities. As one participant stated "Within one specialization, I thought that something had already been included. But then somebody didn't interpret that as being included. So, then we had to make sure to have that conversation." This clarification issue also potentially impacted career advancement trajectories. One concern was that participants indicated that they did not recognize where they could seek funding for their research or where they fit within traditional structures. Many comments along this line included statements along the lines of, "So how do we change the funding infrastructure?" And there were several other

comments regarding how to find funding sources for and difficulties with funding interdisciplinary research.

**Collaborative practicalities.**   Finding and establishing fruitful collaborative teams is a critical element of interdisciplinary research. Therefore, taking an interdisciplinary research perspective often means considering and seeking collaborators outside of typical networks or even home institutions. This lack of immediate support caused some researchers to feel isolated or not know how to begin creating such a network. As one participant stated, "I did not previously consider intentionally reaching out to others to work on projects with, and this workshop showed me that doing so is a good and helpful idea." Another participant stated,

> I think if you're asking disciplinary questions (even questions about learning), it's always going to be difficult to foster distant collaborations. If the problem spans fields (e.g., climate change, the solution for which requires science, policy, etc.), then the distant collaboration seems more natural or necessary.

## Workshop participant's perceived cohesion

During the workshop, several participants, including those who are well-established researchers in their discipline, expressed a sense of professional isolation. Therefore, we formally assessed participant's senses of perceived cohesion within both their self-identified professional communities and their home academic department after the workshop. The self-identified professional communities of the participants included the International Society of the Learning Sciences and the National Association for Research in Science Teaching. Twenty-four out of thirty-three participants complete the survey questions on perceived cohesion. These participants reported a high sense of belonging to their professional community ($M = 4.7$, $SD = 1.1$) but not their department ($M = 3.3$, $SD = .87$, $t(24) = 5.5$, $p < .001$, $d = 1.4$, Fig 1). Conversely, the sense of morale associated with the department ($M = 4.2$, $SD = 1.2$) is much higher than that for the professional community ($M = 3.4$, $SD = .81$, $t(24) = 3.0$, $p < 0.01$, $d = 0.8$, Fig 1).

## Meaningful participant networks

To measure the degree of transformation from the workshop, we determined the extent to which workshop participants forged meaningful networks during the workshop. A visual comparison of the social networks for the participants revealed that the network after the workshop (Fig 2B) was more compact and contained more connections (72 connections) than before the workshop (58 connections, Fig 2A). This observation is supported by the average clustering coefficient of the network which decreased from 0.26 before the workshop to 0.21 after the workshop. Prior to the workshop, participant networks seem to be largely dependent on their self-identified primary area of research (DBER or LS indicated by color in Fig 2), forming discrete disciplinary clusters. After the workshop, these clusters, although identifiable, seem more interwoven as participants formed additional or new cross-disciplinary connections. For example, participant S, a biology education researcher (DBER), formed new connections with participants X, Z, and AA from the learning sciences. Participant Z (LS) also reported forming a new connection with participant Y, a physics education researcher (DBER). Additionally, participant E, a biology education researcher (DBER), formed a new connection with participant K from the learning sciences. These qualitative observations are supported by the greater number of new cross-disciplinary connections (eight) as compared to the number of new

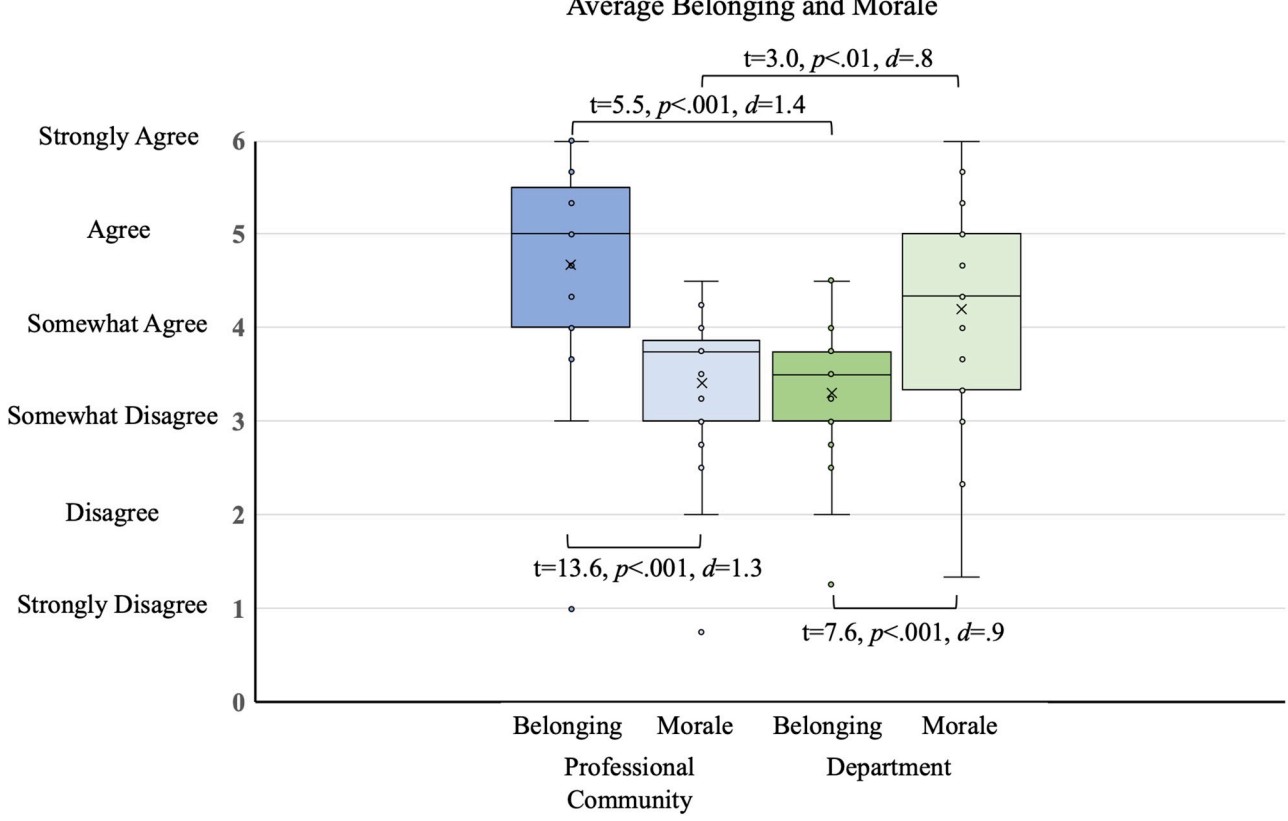

**Fig 1. Sense of belonging and morale in a professional community (blue) and department (green).** 'X' indicates the average.

intradisciplinary connections (three for LS to LS, three for DBER to DBER). The network analysis also revealed that one participant (Y) became a part of the network after the workshop although they were only connected to one person. Other participants appeared to be on the fringes of the network (e.g., participants AB and M) both before and after the workshop.

## Discussion

Interdisciplinary research between LSs and DBERs can facilitate the development of a more nuanced understandings about learning and teaching science. However, we must find strategies to help researchers cross borders and overcome disciplinary differences and siloing to forge interdisciplinary collaborations. Our workshop promoted transformation by purposefully exposing participants to differences in interdisciplinary research perspectives and allowed them time to identify, coordinate, and reflect on these differences along with implications for practice.

Workshop participants reported three primary barriers to LS-DBER interdisciplinary research: *disciplinary differences*, *professional integration*, and *collaborative practicalities*. These barriers were voiced by participants across experience levels, genders, and geographic regions. The identified barriers reflected participant challenges associated with crossing borders. Disciplinary cultural differences require participants to coordinate and consider how to collaborate with people who may use different terminology, practices, and definitions of evidence [13, 16–18]. As suggested by our participants during reflection, developing open communication channels across disciplines can begin to identify and build shared understandings and shared

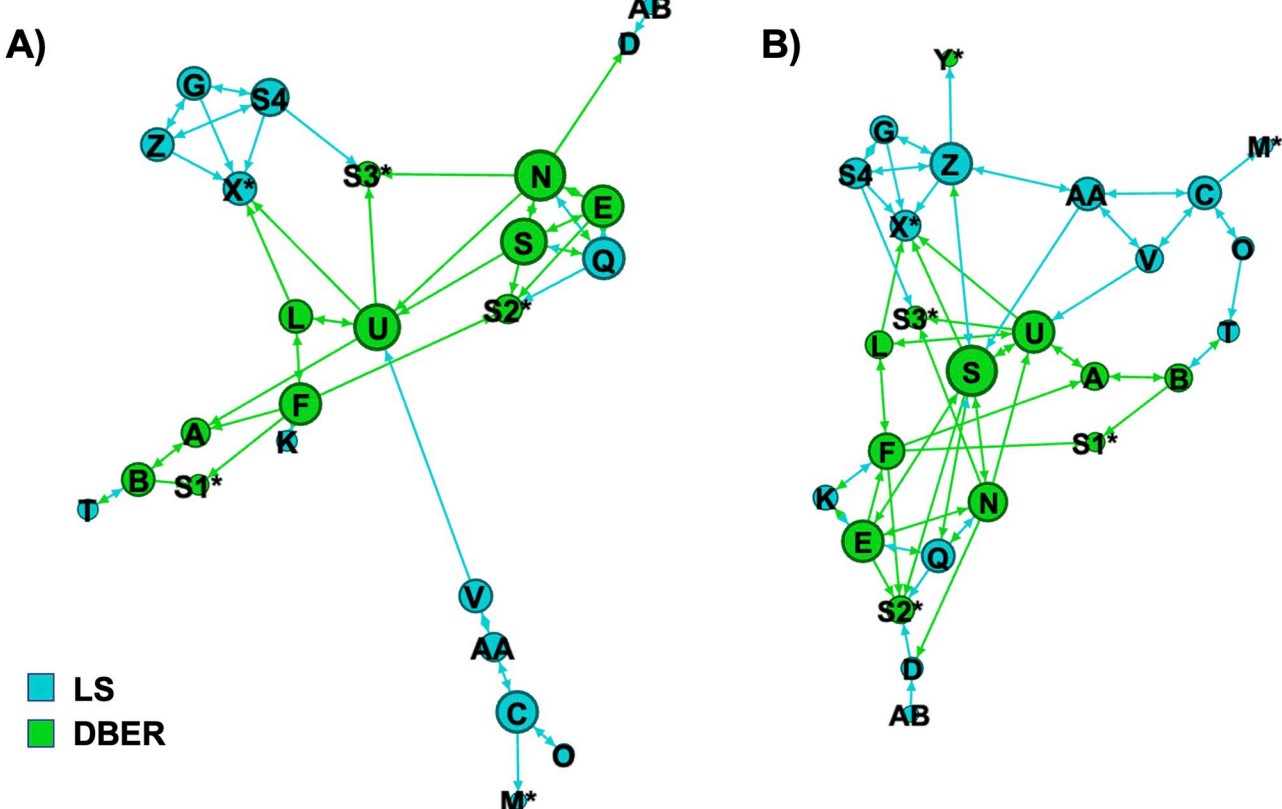

**Fig 2. Network analysis of self-reported contacts between participants A) before and B) after the workshop.** The colors of nodes represent participants' self-reported primary disciplinary identity. L1-L4 are leaders of the workshop, S1-S4 are speakers who also participated in workshop activities. An asterisk indicates that participants did not respond to the networking survey.

language to normalize accepted, rigorous methodological interdisciplinary practices [15, 31]. In addition to workshops like ours, we can provide a space for communication through activities such as the formation of interdisciplinary research Blogs or research journals that solicit contributions from members of both LS and DBER fields.

Initial trends indicated our participants' networks were largely siloed by disciplinary identity and participants struggled with feelings of isolation and not knowing how to reach out to others to establish collaborations [15, 20, 23, 32]. During our workshop, we explicitly created an environment to facilitate transcending disciplinary siloing and encouraged participants to interact with researchers from other disciplines. During the workshop, participants presented their research and worked in small groups. Individuals discussed shared issues that arose within their careers and departments that intradisciplinary colleagues may not understand or appreciate [e.g., 2]. We noted that some participants on the fringes of the network connections were relatively early in their careers. Due to differences in academic rank, early career researchers may struggle to make connections with more established researchers, regardless of discipline [15]. Still, interactions during the workshop led participants to transform their networks by forging connections across disciplines.

Comments made by some participants that aspired to interdisciplinary research suggested that they saw themselves as "disciplinary outcasts" with one foot in each culture [14, 19]. This identified lack of connection to a single discipline can impact perceived cohesion [20, 22, 37],

in turn potentially impacting retention in academia. According to our study, participants confirmed having a low sense of belonging to their home disciplinary departments. However, they reported strong feelings of morale associated with their departmental affiliation. The stronger feelings of morale could be due to social support provided within the department by peers. However, we found the opposite relationship when we asked participants about their connections to professional communities–with a high sense of belonging and low feelings of morale. These low feelings of morale may be due to the professional communities seeming further away and more distant than their own department. The overall differences in the two perceived cohesion factors in the different settings may reflect participant choices. There is more autonomy when selecting which professional community to join, whereas choice in academic positions is based on external restrictions. For example, faculty position availability is often dictated by funding or turnover within the professoriate. Thus, participants may feel a greater sense of belonging to their self-selected professional organizations which better reflect their research interests.

Evidence from our workshop highlights a need to initiate the formation of an interdisciplinary community of practice [32, 40] to combat identified barriers by providing members with a space where they feel a sense of belonging, build self-identity, and engage in the learning mechanisms of identification, coordination, reflection, and transformation associated with cultural border crossing. Communities of practice can also support faculty as they engage in potentially risky interdisciplinary research practice. Building and maintaining such a support network can be accomplished through creating sustainable communication practices, such as interdisciplinary brown-bag lunches, Blogs, or research journals. An interdisciplinary community of practice is critical for supporting professional identity development, changing culture, and establishing clear expectations for interdisciplinary research practices. Our findings offer insights to the development of interdisciplinary guidelines on research expectations, dissemination outlets, and professional milestones to support junior researchers in crossing borders and becoming interdisciplinary researchers.

## Limitations

One limitation of this work is that it focused on one relatively small group of people associated with two disciplines, LS and DBER. Our current data set does not permit investigation of additional factors that influenced identified themes such as whether feelings of isolation within a discipline exacerbate feelings of isolation resulting from belonging to a minority group. Our study takes an important initial step forward towards generating empirical evidence regarding barriers to interdisciplinary research between LSs and DBERs and generating evidence-based strategies for fostering interdisciplinary connections.

## Conclusions

Interdisciplinary collaborations between LSs and DBERs have the potential to be synergistic endeavors that provide insights into the field of undergraduate science education research [3, 4, 6]. However, such collaborations face barriers to fruition due to their interdisciplinary nature. We qualitatively analyzed participant statements during a workshop designed to have participants recognize and develop methods for surmounting the barriers to interdisciplinary research. Three themes emerged as barriers to participation in LS/DBER interdisciplinary research: disciplinary differences, professional integration and collaborative practicalities. Additionally, participants reported a lack of belonging to their department, but a greater sense of belonging to external professional communities. Participants' reports of collaborative connections show that the workshop successfully facilitated the formation of new connections.

We, therefore, suggest that additional meetings geared specifically for facilitating interdisciplinary collaboration would be beneficial in aiding with professional integration and the need to recognize and overcome disciplinary differences. These meetings do not have to take the form of a workshop and could involve people working together to accomplish a goal in a community of practice, or regular lunch and learn meetings supplemented with a blog to facilitate communication of different disciplinary perspectives. To promote interdisciplinary research, we propose the following strategies: 1) expand research on barriers to performing interdisciplinary research; 2) identify factors influencing the impact of these identified barriers so that strategies for lowering these barriers can be implemented; 3) form an interdisciplinary community of practice where researchers can share strategies for overcoming these barriers and derive support. These steps could increase cross-talk between disparate disciplines, decrease feelings of isolation, and promote interdisciplinary research.

## Supporting information

**S1 Data.**
(XLSX)

## Author Contributions

**Conceptualization:** Kristy L. Daniel, Anita Schuchardt, Melanie E. Peffer.

**Data curation:** Anita Schuchardt, Melanie E. Peffer.

**Formal analysis:** Kristy L. Daniel, Myra McConnell, Anita Schuchardt, Melanie E. Peffer.

**Funding acquisition:** Kristy L. Daniel, Anita Schuchardt, Melanie E. Peffer.

**Investigation:** Kristy L. Daniel, Myra McConnell, Anita Schuchardt, Melanie E. Peffer.

**Methodology:** Kristy L. Daniel, Myra McConnell, Anita Schuchardt, Melanie E. Peffer.

**Project administration:** Anita Schuchardt, Melanie E. Peffer.

**Resources:** Kristy L. Daniel, Anita Schuchardt, Melanie E. Peffer.

**Software:** Kristy L. Daniel.

**Supervision:** Kristy L. Daniel, Anita Schuchardt.

**Validation:** Kristy L. Daniel, Myra McConnell.

**Visualization:** Kristy L. Daniel, Anita Schuchardt.

**Writing – original draft:** Kristy L. Daniel, Myra McConnell, Anita Schuchardt, Melanie E. Peffer.

**Writing – review & editing:** Kristy L. Daniel, Myra McConnell, Anita Schuchardt, Melanie E. Peffer.

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
