## [Decision Letter · Decision Letter 0]

7 Feb 2022

PONE-D-21-38247Challenges facing interdisciplinary researchers: Findings from a professional development workshopPLOS ONE

Dear Dr. Peffer,

Thank you for submitting your manuscript to PLOS ONE. After careful consideration, we feel that it has merit but does not fully meet PLOS ONE’s publication criteria as it currently stands. Therefore, we invite you to submit a revised version of the manuscript that addresses the points raised during the review process.

We look forward to receiving your revised manuscript.

Kind regards,

Mehmet Serkan Kirgiz

Academic Editor

PLOS ONE

Journal Requirements:

Additional Editor Comments:

Dear Dr. Peffer:

Ref: Challenges facing interdisciplinary researchers: Findings from a professional development workshop

Our reviewers have considered your paper and have recommended publication in Plos One, after minor revision of it you will make. I am pleased to accept your paper after you could revise it to the comments of Reviewer 1. The reviewer comments are included at the bottom of this letter, along with those of the editor who coordinated the review of your paper. To guide you, I have attached the comments of reviewers below.

Thank you considering the Plos One as your publishing partner.

Best regards,

Professor Dr. M. Serkan KIRGIZ

Academic Editor of Plos One

Reviewers' comments:

Reviewer's Responses to Questions

**Comments to the Author**

1. Is the manuscript technically sound, and do the data support the conclusions?

Reviewer #1: Yes

Reviewer #2: Yes

2. Has the statistical analysis been performed appropriately and rigorously? 

Reviewer #1: N/A

Reviewer #2: Yes

3. Have the authors made all data underlying the findings in their manuscript fully available?

Reviewer #1: No

Reviewer #2: Yes

4. Is the manuscript presented in an intelligible fashion and written in standard English?

Reviewer #1: Yes

Reviewer #2: Yes

5. Review Comments to the Author

Reviewer #1: This paper was well written and written in an engaging style. Although it was written with reference to interdisciplinary research involving DBERs and LSs researchers it is relevant to interdisciplinary research more generally across academic disciplines. It highlighted well the benefits of interdisciplinary research and the barriers to carrying it out , both cultural and structural. I felt , however, that the theoretical framework was a little repetitive and could have been tightened up and the diagram 1 was unnecessary.

Reviewer #2: Authors presented a nice work with live study and statistical analysis of the interdisciplinary research. Author have discussed a case studies of workshop and presented the statistical record.

Data was collected from three sources: surveys of participants, field notes from workshop observations, and artifacts from the workshop including chat transcripts and the Google slides. Additionally, during the workshop, video-recording occurred for full group and breakout discussions, activities, and poster presentations.

6. PLOS authors have the option to publish the peer review history of their article (what does this mean?). If published, this will include your full peer review and any attached files.

Reviewer #1: No

Reviewer #2: No

---

## [Author Response · Author response to Decision Letter 0]

1 Apr 2022

Dear Dr. Serkan,

Thank you for the opportunity to revise our manuscript entitled, Challenges facing interdisciplinary researchers: Findings from a professional development workshop. We appreciate you taking the time to review our work and find suitable reviewers. We have reviewed the provided feedback from the two reviewers and have made the following revisions [See notes in Blue]:

Comments to the Author

1. Is the manuscript technically sound, and do the data support the conclusions?

Reviewer #1: Yes

Reviewer #2: Yes

[We did not make any changes related to this issue]

2. Has the statistical analysis been performed appropriately and rigorously? 

Reviewer #1: N/A

Reviewer #2: Yes

[We did not make any changes related to this issue]

3. Have the authors made all data underlying the findings in their manuscript fully available?

Reviewer #1: No

Reviewer #2: Yes

[We have included a blinded data set that can be made available as needed for readers]

4. Is the manuscript presented in an intelligible fashion and written in standard English?

Reviewer #1: Yes

Reviewer #2: Yes

[We did not make any changes related to this issue]

5. Review Comments to the Author

Reviewer #1: This paper was well written and written in an engaging style. Although it was written with reference to interdisciplinary research involving DBERs and LSs researchers it is relevant to interdisciplinary research more generally across academic disciplines. It highlighted well the benefits of interdisciplinary research and the barriers to carrying it out , both cultural and structural. I felt , however, that the theoretical framework was a little repetitive and could have been tightened up and the diagram 1 was unnecessary.

Reviewer #2: Authors presented a nice work with live study and statistical analysis of the interdisciplinary research. Author have discussed a case studies of workshop and presented the statistical record.

Data was collected from three sources: surveys of participants, field notes from workshop observations, and artifacts from the workshop including chat transcripts and the Google slides. Additionally, during the workshop, video-recording occurred for full group and breakout discussions, activities, and poster presentations.

[We revised our theoretical framework to reduce redundancy and removed Figure 1. We also reviewed our references for completion and corrected formatting requirements where needed.]

Thank you again for considering our revised manuscript and we look forward to publishing our work in PLOS ONE.

---

## [Editor Report · Decision Letter 1]

5 Apr 2022

Challenges facing interdisciplinary researchers: Findings from a professional development workshop

PONE-D-21-38247R1

Dear Dr. Peffer,

We’re pleased to inform you that your manuscript has been judged scientifically suitable for publication and will be formally accepted for publication once it meets all outstanding technical requirements.

Kind regards,

Professor Dr. Mehmet Serkan Kirgiz

Academic Editor

PLOS ONE
---

## [Editor Report · Acceptance letter]

11 Apr 2022

PONE-D-21-38247R1 

Challenges facing interdisciplinary researchers: Findings from a professional development workshop 

Dear Dr. Peffer:

I'm pleased to inform you that your manuscript has been deemed suitable for publication in PLOS ONE. Congratulations! Your manuscript is now with our production department. 

Kind regards, 

on behalf of

Professor Dr. Mehmet Serkan Kirgiz 

Academic Editor

PLOS ONE